# Time/Accuracy Tradeoffs for Learning a ReLU with respect to Gaussian Marginals

**Surbhi Goel**
Department of Computer Science
University of Texas at Austin
surbhi@cs.utexas.edu

**Sushrut Karmalkar**
Department of Computer Science
University of Texas at Austin
sushrutk@cs.utexas.edu

**Adam R. Klivans**
Department of Computer Science
University of Texas at Austin
klivans@cs.utexas.edu

## Abstract

We consider the problem of computing the best-fitting ReLU with respect to square-loss on a training set when the examples have been drawn according to a spherical Gaussian distribution (the labels can be arbitrary). Let opt $< 1$ be the population loss of the best-fitting ReLU. We prove:

- Finding a ReLU with square-loss opt $+ \epsilon$ is as hard as the problem of learning sparse parities with noise, widely thought to be computationally intractable. This is the first hardness result for learning a ReLU with respect to Gaussian marginals, and our results imply –*unconditionally*– that gradient descent cannot converge to the global minimum in polynomial time.
- There exists an efficient approximation algorithm for finding the best-fitting ReLU that achieves error $O(\mathsf{opt}^{2/3})$. The algorithm uses a novel reduction to noisy halfspace learning with respect to $0/1$ loss.

Prior work due to Soltanolkotabi [Sol17] showed that gradient descent *can* find the best-fitting ReLU with respect to Gaussian marginals, if the training set is *exactly* labeled by a ReLU.

## 1 Introduction

A Rectified Linear Unit (ReLU) is a function parameterized by a weight vector $\mathbf{w} \in \mathbb{R}^d$ that maps $\mathbb{R}^d \to \mathbb{R}$ as follows: $\mathsf{ReLU}_{\mathbf{w}}(\mathbf{x}) = \max(0, \mathbf{w} \cdot \mathbf{x})$. ReLUs are now the nonlinearity of choice in modern deep networks. The computational complexity of learning simple neural networks that use the ReLU activation is an intensely studied area, and many positive results rely on assuming that the marginal distribution on the examples is a spherical Gaussian [ZYWG19, GLM18, ZSJ$^+$17, MR18]. Recent work due to Soltanolkotabi [Sol17] shows that gradient descent will learn a single ReLU in polynomial time, if the marginal distribution is Gaussian (see also [BG17]). His result, however, requires that the training set is *noiseless*; i.e., there is a ReLU that correctly classifies all elements of the training set.

Here we consider the more realistic scenario of empirical risk minimization or learning a ReLU with noise (often referred to as *agnostically* learning a ReLU). We assume that a learner has access to a training set from a joint distribution $\mathcal{D}$ on $\mathbb{R}^d \times \mathbb{R}$ where the marginal distribution on $\mathbb{R}^d$ is Gaussian but the distribution on the labels can be arbitrary within $[0, 1]$. We define

$\mathsf{opt} = \min_{w, \|w\| \leq 1} \mathbb{E}_{x, y \sim \mathcal{D}}[(\mathsf{ReLU}_w(x) - y)^2]$, and the goal is to output a function of the form $\max(0, \mathbf{w} \cdot \mathbf{x})$ with square-loss at most $\mathsf{opt} + \epsilon$.

## 1.1 Our Results

Our main results give a trade-off between the accuracy of the output hypothesis and the running time of the algorithm. We give the first evidence that there is no polynomial-time algorithm for finding a ReLU with error $\mathsf{opt} + \epsilon$, even when the marginal distribution is Gaussian:

**Theorem 1** (Informal version of Theorem 3). *Assuming hardness of the problem of learning sparse parities with noise, any algorithm for finding a ReLU on data drawn from a distribution with Gaussian marginals that has error at most $\mathsf{opt} + \epsilon$ runs in time $d^{\Omega(\log(1/\epsilon))}$.*

Since gradient descent is known to be a *statistical-query* algorithm (see Section 4), a consequence of Theorem 1 is the following:

**Corollary 1.** *Gradient descent fails to converge to the global minimum for learning the best-fitting ReLU with respect to square-loss in polynomial time, even when the marginals are Gaussian.*

This above corollary is unconditional (i.e. does not rely on any hardness assumptions) and shows the necessity of the realizable/noiseless setting in the work of Soltanolkotabi [Sol17] and Brutzkus and Globerson [BG17]. We also give the first approximation algorithm for finding the best-fitting ReLU with respect to Gaussian marginals:

**Theorem 2** (Informal version of Theorem 5). *There exists a polynomial-time algorithm for finding a ReLU with error $O(\mathsf{opt}^{2/3}) + \epsilon$.*

The above result uses a novel reduction from learning a ReLU to the problem of learning a halfspace with respect to $0/1$ loss. We note that the problem of finding a ReLU with error $O(\mathsf{opt}) + \epsilon$ remains an outstanding open problem.

## 1.2 Our Techniques

**Hardness Result.** For our hardness result, we follow the same approach as Klivans and Kothari [KK14] who gave a reduction from learning sparse parity with noise to the problem of agnostically learning *halfspaces* with respect to Gaussian distributions. The idea is to embed examples drawn from $\{-1, 1\}^d$ into $\mathbb{R}^d$ by multiplying each coordinate with a random draw from a half-normal distribution. The key technical component in their result is a correlation lemma showing that for a parity function on variables indicated by index set $S$, the majority function on the same index set is weakly correlated with a *Gaussian lift* of the parity function on $S$.

In our work we must overcome two technical difficulties. First, in the Klivans and Kothari result, it is obvious that for distributions induced by learning sparse parity with noise, the best fitting majority function will be the one that is defined on inputs specified by $S$. In our setting with respect to ReLUs, however, the constant function $1/2$ will have square-loss $1/4$, and this may be much lower than the square-loss of any function of the form $\max(0, \mathbf{w} \cdot \mathbf{x})$. Thus, we need to prove the existence of a gap between the correlation of ReLUs with random noise (see Claim 3) versus the correlation of ReLUs with parity (see Claim 4).

Second, Klivans and Kothari use known formulas on the discrete Fourier coefficients of the majority function and an application of the central limit theorem to analyze how much the best-fitting majority correlates with the Gaussian lift of parity. No such bounds are known, however, for the ReLU function. As such we must perform a (somewhat involved) analysis of the ReLU function's Hermite expansion in order to obtain quantitative correlation bounds.

**Approximation Algorithm.** For our polynomial-time algorithm that outputs a ReLU with error $O(\mathsf{opt}^{2/3}) + \epsilon$, we apply a novel reduction to agnostically learning halfspaces. We give a simple transformation on the training set to a Boolean learning problem and show that the weight vector $\mathbf{w}$ corresponding to the best fitting *halfspace* on this transformed data set is not too far from the weight vector corresponding to the best fitting ReLU. We can then apply recent work for agnostically learning halfspaces with respect to Gaussians that have constant-factor approximation error guarantees. The exponent $2/3$ appears due to the use of an averaging argument (see Section 5).

## 1.3 Related Work

Several recent works have proved hardness results for finding the best-fitting ReLU with respect to square loss (equivalently, agnostically learning a ReLU with respect to square loss). Results showing NP-hardness (e.g., [MR18, BDL18]) use marginal distributions that encode hard combinatorial problems. The resulting marginals are far from Gaussian. Work due to Goel et al. [GKKT17] uses a reduction from sparse parity with noise but only obtains hardness results for learning with respect to discrete distributions (uniform on $\{0, 1\}^d$).

Using parity functions as a source of hardness for learning deep networks has been explored recently by Shalev-Shwartz et. al. [SSSS17] and Abbe and Sandon [AS18]. Their results, however, do not address the complexity of learning a *single* ReLU or consider the case of Gaussian marginals. Shamir [Sha18] proved that gradient descent fails to learn certain classes of neural networks with respect to Gaussian marginals, but these results do not apply to learning a single ReLU [VW19].

In terms of positive results for learning a ReLU, work due to Kalai and Sastry [KS09] (and follow-up work [KKKS11]) gave the first efficient algorithm for learning any generalized linear model (GLM) that is monotone and Lipschitz, a class that includes ReLUs. Their algorithms work for *any* distribution and can tolerate bounded, mean-zero and additive noise. Soltanolkotabi [Sol17] and Brutzkus and Globerson [BG17] were the first to prove that gradient descent converges to the unknown ReLU in polynomial time with respect to Gaussian marginals as long as the labels have *no noise*. Other works for learning one-layer ReLU networks with respect to Gaussian marginals or marginals with milder distribution assumptions [ZYWG19, GLM18, ZSJ$^+$17, GKLW19, GKM18, MR18] also assume a noiseless training set or training set with mean-zero i.i.d. (typically sub-Gaussian) noise. This is in contrast to the setting here (agnostic learning), where we assume nothing about the noise model.

There are several works for the related (but different) problem of agnostically learning *halfspaces* with respect to Gaussian marginals [KKMS08, ABL14, Zha18, DKS18]. While agnostically learning ReLUs may seem like an easier problem than agnostically learning halfspaces (at first glance the learner sees "more information" from the ReLU's real-valued labels), the quantitative relationship between the two problems is still open. In the halfspace setting, we can assume without loss of generality that an adversary has flipped an opt fraction of the labels. In contrast, in the setting with ReLUs and square loss, it is possible for the adversary to corrupt *every* label.

## 2 Preliminaries

Define $\mathsf{ReLU}(a) = \max(0, a)$ and the set of functions $\mathcal{C}_{\mathsf{ReLU}} := \{\mathsf{ReLU}_\mathbf{w} | \mathbf{w} \in \mathbb{R}^d, \|\mathbf{w}\|_2 \le 1\}$ where $\mathsf{ReLU}_\mathbf{w}(\mathbf{x}) = \max(0, \mathbf{w} \cdot \mathbf{x})$. Define $\mathsf{sign}(a)$ to be 1 if $a \ge 0$ and -1 otherwise. Let $\mathsf{err}_\mathcal{D}(h) := \mathbb{E}_{(\mathbf{x},y) \sim \mathcal{D}}[(h(\mathbf{x}) - y)^2]$, Also define $\mathsf{opt}_\mathcal{D}(\mathcal{C}) = \min_{c \in \mathcal{C}} \mathsf{err}_\mathcal{D}(c)$ to be the error of the best-fitting $c \in \mathcal{C}$ for distribution $\mathcal{D}$. We will use $\mathbf{x}_{-i}$ to denote the vector $\mathbf{x}$ restricted to the indices except $i$. The 'half-normal' distribution will refer to the standard normal distribution truncated to $\mathbb{R}^{\ge 0}$. We will use $n$ and its subscripted versions to denote natural numbers unless otherwise stated. In this paper, we will suppress the confidence parameter $\delta$, since one can use standard techniques to amplify the probability of success of our learning algorithms.

**Agnostic learning.** The model of learning we work with in the paper is the agnostic model of learning. In this model the labels are allowed to be arbitrary and the task of the learner is to output a hypothesis within an $\epsilon$ error of the optimal. More formally,

**Definition 1.** *A class $\mathcal{C}$ is said to be agnostically learnable in time $t$ over the Gaussian distribution to error $\epsilon$ if there exists an algorithm $\mathcal{A}$ such that for any distribution $\mathcal{D}$ on $X \times Y$ with the marginal on $X$ being Gaussian, $\mathcal{A}$ uses at most $t$ draws from $\mathcal{D}$, runs in time at most $t$, and outputs a hypothesis $h \in \mathcal{C}$ such that $\mathsf{err}_\mathcal{D}(h) \le \mathsf{opt}_\mathcal{D}(\mathcal{C}) + \epsilon$.*

We assume that $\mathcal{A}$ succeeds with constant probability. Note that the algorithm above outputs the "best-fitting" $c \in \mathcal{C}$ with respect to $\mathcal{D}$ up to an additive $\epsilon$. We will denote $\widehat{\mathsf{err}}_\mathcal{S}(h)$ to be the empirical error of $h$ over samples $\mathcal{S}$.

**Learning Sparse Parities with Noise.** In this work we will show that agnostically learning $\mathcal{C}_{\mathsf{ReLU}}$ over the Gaussian distribution is as hard as the problem of learning sparse parities with noise over the uniform distribution on the hypercube.

**Definition 2** ($k$-SLPN). *Given access to samples drawn from the uniform distribution over $\{\pm 1\}^d$ and target function $y$ being the parity function over an unknown set $S \subseteq [d]$ of size $k$, the problem of learning sparse parities with noise is the problem of recovering the set $S$ given access to noisy labels where the label is flipped with probability $\eta$.*

Learning sparse parities with noise is generally considered to be a computationally hard problem and has been used to give hardness results for both supervised [GKKT17] and unsupervised learning problems [BGS14]. The current best known algorithm for solving sparse parities with constant noise rate is due to Valiant [Val15] and runs in time $\approx d^{0.8k}$.

**Assumption 1.** *Any algorithm for solving $k$-SLPN up to constant error must run in time $d^{\Omega(k)}$.*

**Gaussian Lift of a Function**   Our reduction will require the following definition of a Gaussian lift of a boolean function from [KK14].

**Definition 3** (Gaussian lift [KK14]). *The Gaussian lift of a function $f : \{\pm 1\}^d \to \mathbb{R}$ is the function $f^\gamma : \mathbb{R}^d \to \mathbb{R}$ such that for any $x \in \mathbb{R}^d$, $f^\gamma(x) = f(\mathsf{sign}(x_1), \dots, \mathsf{sign}(x_d))$.*

**Hermite Analysis and Gaussian Density**   We will assume that the marginal over our samples $\mathbf{x}$ is the standard normal distribution $N(0, I_d)$. This implies that $\mathbf{w} \cdot \mathbf{x}$ for a vector $\mathbf{w}$ is distributed as $N(0, \|\mathbf{w}\|^2)$. We recall the basics of Hermite analysis. We say a function $f : \mathbb{R} \to \mathbb{R}$ is square integrable if $\mathbb{E}_{N(0,1)}[f^2] < \infty$. For any square integrable function $f$ define its Hermite expansion as $f(x) = \sum_{i=0}^\infty \widehat{f}_i \bar{H}_i(x)$ where $\bar{H}_i(x) = \frac{H_i(x)}{\sqrt{i!}}$ are the normalized Hermite polynomials, and $H_i$ the unnormalized (probabilists) Hermite polynomials. The normalized Hermite polynomials form an orthonormal basis with respect to the univariate standard normal distribution ($\mathbb{E}[\bar{H}_i(x)\bar{H}_j(x)] = \delta_{ij}$). The associated inner product for square integrable functions $f, g : \mathbb{R} \to \mathbb{R}$ is defined as $\langle f, g \rangle := \mathbb{E}_{x \sim N(0,1)}[f(x)g(x)]$. Each coefficient $\widehat{f}_i$ in the expansion of $f(x)$ satisfies $\widehat{f}_i = E_{x \sim N(0,1)}[f(x)\bar{H}_i(x)]$. We will need the following facts about Hermite polynomials.

**Fact 1.** *For all $m \geq 0$, $H_{2m+1}(0) = 0$ and $H_{2m} = (-1)^m \frac{(2m)!}{m! 2^m}$.*

**Fact 2** ([KKMS08]). *$\widehat{\mathsf{sign}}_0 = 0$ and for $i \geq 1$, $\widehat{\mathsf{sign}}_i = \sqrt{\frac{2}{\pi i!}} H_{i-1}(0)$.*

# 3   Hardness of Learning ReLU

In this section, we will show that if there is an algorithm that agnostically learns a ReLU in polynomial time, then there is an algorithm for learning sparse parities with noise in time $d^{o(k)}$, violating Assumption 1. We follow the approach of [KK14]. Let $\chi_S$ be an unknown parity for some $S \subseteq [d]$. We will show that there is an unbiased ReLU that is correlated with the Gaussian lift of the unknown sparse parity function. Notice that dropping a coordinate $j \in S$ from the input samples makes the labels of the resulting training set totally independent from the input. In contrast, dropping $j \notin S$ results in a training set that is still labeled by a noisy parity. Therefore, we can use an agnostic learner for ReLUs to detect a correlated ReLU and distinguish between the two cases. This allows us to identify the variables in $S$ one by one.

We formalize the above approach by first proving the following key property,

**Lemma 1** (ReLU Correlation Lemma). *Let $\chi_S^\gamma$ denote the Gaussian lift of the parity on variables in $S \subset [d]$. For every $S \subset [d]$ with $|S| \leq k$ and $k = 4l + 2$ for some $l \geq 0$, there exists $\mathsf{ReLU}_{\mathbf{w}_S}$ such that $\langle \mathsf{ReLU}_{\mathbf{w}_S}, \chi_\alpha^\gamma \rangle \geq 2^{-O(k)}$ where $\mathsf{ReLU}_{\mathbf{w}_S}$ only depends on variables in $S$.*

*Proof.* Let $\mathbf{w}_S = \frac{1}{\sqrt{2\pi k}} \sum_{i \in S} \mathbf{e}^{(i)}$ where $\mathbf{e}^{(i)}$ is 1 at coordinate $i$ and 0 everywhere else. We will show that

$$\langle \mathsf{ReLU}_{\mathbf{w}_S}, \chi_S^\gamma \rangle = \frac{1}{\sqrt{2\pi}} \mathbb{E}_{\mathbf{z} \sim \mathcal{N}(0, \mathbf{I}_d)} \left[ \mathsf{ReLU}\left( \frac{\sum_{i \in S} z_i}{\sqrt{k}} \right) \left( \prod_{i \in S} \mathsf{sign}(z_i) \right) \right] \geq 2^{-O(k)}.$$

Let $\widehat{\text{sign}}_n$ and $\widehat{\text{ReLU}}_n$ denote the degree $n$ Hermite coefficients of the sign function and ReLU function respectively. It is easy to see that the Hermite expansion of the Gaussian lift of a parity supported on $S$ is,

$$\chi_S^\gamma(\mathbf{z}) = \prod_{i \in S} \text{sign}(z_i) = \prod_{i \in S}\left(\sum_{n=0}^\infty \widehat{\text{sign}}_n \bar{H}_n(z_i)\right) = \sum_{n_1,\dots,n_k} \prod_{i \in S} \widehat{\text{sign}}_{n_i} \bar{H}_{n_i}(z_i) \tag{1}$$

In order to finish the proof of Lemma 1 we will need the expansion of $\text{ReLU}\left(\frac{\sum_i z_i}{\sqrt{k}}\right)$ in terms of products of univariate Hermite polynomials. Toward this end we establish the following two claims (see proofs in the supplemental).

**Claim 1** (Hermite expansion: univariate ReLU). $\widehat{\text{ReLU}}_0 = 1/\sqrt{2\pi}$, $\widehat{\text{ReLU}}_1 = 1/2$ and for $i \geq 2$, $\widehat{\text{ReLU}}_i = \frac{1}{\sqrt{2\pi i!}}(H_i(0) + iH_{i-2}(0))$.

**Claim 2** (Hermite expansion: multivariate ReLU). *For any $S \subseteq [d]$ with $|S| = k$,*

$$\text{ReLU}\left(\frac{\sum_{i \in S} z_i}{\sqrt{k}}\right) = \sum_{n=0}^\infty \frac{\widehat{\text{ReLU}}_n}{k^{n/2}} \cdot \sum_{n_1+\dots+n_k=n}\left(\frac{n!}{n_1!\cdots n_k!}\right)^{1/2} \prod_{j=1}^k \bar{H}_{n_j}(z_j)$$

Combining Equation 1 and Claim 2 now yields,

$$\mathbb{E}_{\mathbf{z} \sim \mathcal{N}(0,\mathbf{I}_d)}\left[\text{ReLU}\left(\frac{\sum_{i \in S} z_i}{\sqrt{k}}\right) \cdot \prod_{i \in S}\text{sign}(z_i)\right]$$

$$= \mathbb{E}_{\mathbf{z} \sim \mathcal{N}(0,\mathbf{I}_d)}\left[\left(\sum_{n=0}^\infty \frac{\widehat{\text{ReLU}}_n}{k^{n/2}} \sum_{n_1+\dots+n_k=n}\left(\frac{n!}{n_1!\cdots n_k!}\right)^{1/2}\prod_{i=1}^k \bar{H}_{n_i}(z_i)\right)\right.$$

$$\left.\times \left(\sum_{m_1,\dots,m_k}\prod_{j=1}^k \widehat{\text{sign}}_{m_j}\bar{H}_{m_j}(z_j)\right)\right]$$

$$= \sum_{n=0}^\infty \frac{\widehat{\text{ReLU}}_n}{k^{n/2}} \sum_{n_1+\dots+n_k=n}\sum_{m_1,\dots,m_k}\left(\frac{n!}{n_1!\cdots n_k!}\right)^{1/2}\prod_{i=1}^k \widehat{\text{sign}}_{m_i}\mathbb{E}[\bar{H}_{n_i}(z_i)\bar{H}_{m_i}(z_i)]$$

$$= \sum_{n=0}^\infty \frac{\widehat{\text{ReLU}}_n}{k^{n/2}} \sum_{n_1+\dots+n_k=n}\left(\frac{n!}{n_1!\cdots n_k!}\right)^{1/2}\prod_{i=1}^k \widehat{\text{sign}}_{n_i}$$

From Fact 2 and Claim 1 we see that $\widehat{\text{sign}}_{2m} = 0$ and $\widehat{\text{ReLU}}_{2m+1} = 0$ for $m \geq 1$. Additionally, since $\widehat{\text{sign}}_0 = 0$ we see that each $n_i \geq 1$. This gives us,

$$\mathbb{E}_{\mathbf{z} \sim \mathcal{N}(0,\mathbf{I}_d)}\left[\text{ReLU}\left(\frac{\sum_{i \in S} z_i}{\sqrt{k}}\right) \cdot \prod_{i \in S}\text{sign}(z_i)\right]$$

$$= \sum_{n=k}^\infty \frac{1}{\sqrt{2\pi n!}k^{n/2}}(H_n(0) + nH_{n-2}(0)) \sum_{\substack{n_1,\dots,n_k \geq 1 \\ n_1+\dots+n_k=n}}\left(\frac{n!}{n_1!\cdots n_k!}\right)^{1/2}\prod_{i=1}^k\sqrt{\frac{2}{\pi n_j!}}H_{n_j-1}(0)$$

$$= \sum_{n=k}^\infty \frac{1}{\sqrt{2\pi}k^{n/2}}(H_n(0) + nH_{n-2}(0)) \sum_{\substack{n_1,\dots,n_k \geq 1 \\ n_1+\dots+n_k=n}}\left(\frac{1}{n_1!\cdots n_k!}\right)^{3/2}\prod_{i=1}^k\sqrt{\frac{2}{\pi}}H_{n_j-1}(0).$$

To finish the proof of Lemma 1, we will look at each term in the outer summation above. Let the term for any fixed $n \geq k$ be denoted by $T_n$. Since $\bar{H}_i(0) = 0$ for odd $i$, observe that $T_n$ is non-zero if and only if $n$ is even and each $n_i = 2n_i' + 1$ for $n_i' \geq 0$. We have

$$T_n = \frac{1}{\sqrt{2\pi}k^{\frac{n}{2}}}\left((-1)^{\frac{n}{2}}\frac{n!}{(n/2)!2^{\frac{n}{2}}} + n(-1)^{\frac{n}{2}-1}\frac{(n-2)!}{(\frac{n}{2}-1)!2^{\frac{n}{2}-1}}\right)$$

$$\times \sum_{\substack{n'_i \geq 0 \\ \sum_j n'_j = \frac{n-k}{2}}} \left( \frac{1}{(2n'_1+1)!\cdots(2n'_k+1)!} \right)^{3/2} \prod_{j=1}^{k} \sqrt{\frac{2}{\pi}} (-1)^{n'_j} \frac{(2n'_j)!}{n'_j! 2^{n'_j}}$$

$$= \frac{(-1)^{\frac{n}{2}-1} n!}{\sqrt{2\pi} k^{\frac{n}{2}} (n/2)! 2^{\frac{n}{2}}} \left( -1 + \frac{n}{n-1} \right) \frac{2^{\frac{k}{2}} (-1)^{\frac{n-k}{2}}}{\pi^{\frac{k}{2}}}$$

$$\times \sum_{\substack{n'_j \geq 0 \\ \sum_j n'_j = \frac{n-k}{2}}} \left( \frac{1}{(2n'_1+1)!\cdots(2n'_k+1)!} \right)^{3/2} \frac{(2n'_1)!\cdots(2n'_k)!}{n'_1!\cdots n'_k!}$$

$$= \frac{(-1)^{1+\frac{k}{2}} n!}{\sqrt{2\pi} k^{\frac{n}{2}} (n-1)(n/2)! 2^{\frac{n-k}{2}} \pi^{\frac{k}{2}}} \sum_{\substack{n'_i \geq 0 \\ \sum_j n'_j = \frac{n-k}{2}}} \left( \frac{1}{(2n'_1+1)!\cdots(2n'_k+1)!} \right)^{3/2} \frac{(2n'_1)!\cdots(2n'_k)!}{n'_1!\cdots n'_k!}$$

Since $k = 4l+2$ (by assumption), $T_n > 0$ for all even $n \geq k$ and equal to 0 for all odd $n$. Thus $\sum_{n=k}^{\infty} T_n > T_k$. Lower bounding $T_k$, we have

$$T_k = \frac{(4l+2)!}{\sqrt{2\pi}(4l+2)^{2l+1}(4l+1)(2l+1)!\pi^{2l+1}}$$

$$\approx \frac{1}{\sqrt{2\pi}(4l+2)^{2l+1}(4l+1)\pi^{2l+1}} \sqrt{2\pi(4l+2)} \left( \frac{4l+2}{e} \right)^{4l+2} \frac{1}{\sqrt{2\pi(2l+1)}} \left( \frac{e}{2l+1} \right)^{2l+1}$$

$$= \frac{1}{\sqrt{\pi}(4l+1)} \left( \frac{2}{e\pi} \right)^{2l+1} = 2^{-O(k)}$$

$\square$

Now we present our main algorithm (Algorithm 1) that reduces learning sparse parities with noise to agnostically learning ReLUs and a proof of its correctness.

---

**Algorithm 1** Learning Sparse Parities with Noise using Agnostic ReLU learner

---

     **Input** Training set $\mathcal{S}$ of $M_1$ samples $(\mathbf{x}^i, y^i)_{i=1}^{M_1}$, validation set $\mathcal{V}$ of $M_2$ samples
         $(\mathbf{x}^i, y^i)_{i=M_1+1}^{M_1+M_2}$, error parameter $\epsilon$, Agnostic ReLU learner $\mathcal{A}$
     **Output** Set of relevant variables $V_{rel}$

1: Set $V_{rel} = \emptyset$
2: Set $\mathcal{S}_1, \ldots, \mathcal{S}_d := \emptyset$
3: **for** $i = 1$ to $M_1 + M_2$ **do**
4:     Draw $n$ independent univariate half Gaussians $g_1, \ldots, g_d$
5:     Construct $\mathbf{x}'$ such that for all $j \in [d]$, $x'_j := g_j x^i_j$ and set $y' = \frac{y^i+1}{2}$
6:     For all $j \in [d]$, if $i \leq M_1$ add $(\mathbf{x}'_{-j}, y')$ to $\mathcal{S}_j$ else to $\mathcal{V}_j$

7: **for** $j \in [d]$ **do**
8:     Run $\mathcal{A}$ on $\mathcal{S}_j$ to obtain hypothesis $h_j$
9:     Compute $\widehat{\text{err}}_{\mathcal{V}_j}(h_j)$
10:     **if** $\widehat{\text{err}}_{\mathcal{V}_j}(h_j) \geq \frac{1}{2} - \frac{1}{4\pi} - \epsilon/4$ **then**
11:         Add $j$ to $V_{rel}$
12: Return $V_{rel}$

---

**Theorem 3.** *If there is an algorithm to agnostically learn unbiased ReLUs on the Gaussian distribution in time and samples $T(d, 1/\epsilon)$, then there is an algorithm to solve $k$-SLPN in time $O\left( \frac{2^{O(k)}}{(1-2\eta)^2} \log(d) \right) + O(d) T\left( d, \frac{2^{O(k)}}{1-2\eta} \right)$ where $\eta$ is the noise rate.*

*In particular, if Assumption 1 is true, then any algorithm for agnostically learning (unbiased) ReLUs on the Gaussian distribution must run in time $d^{\Omega(\log(1/\epsilon))}$.*

*Proof.* Given a set of samples from the $k$-SPLN problem, we claim that Algorithm 1 can recover all indices $j$ belonging to the sparse parity when run with appropriate parameters. We will first show that if a variable is relevant then the error is smaller compared to when it is irrelevant. It is easy to see that $y'$ is $\frac{\prod_{i \in S} \text{sign}(x_i') + 1}{2}$ with probability $1 - \eta$ and $\frac{1 - \prod_{i \in S} \text{sign}(x_i')}{2}$ otherwise. Let $\mathcal{D}_j$ denote the distribution obtained by dropping the $j$th coordinate from the lifted distribution and let $S$ denote the set of active indices of the parity. The proof of the theorem follows from the following claims,

**Claim 3.** *If $j \in S$ then for all $w$,* $\text{err}_{\mathcal{D}_j}(\text{ReLU}_w) = \frac{\|w\|^2}{2} - \frac{\|w\|}{\sqrt{2\pi}} + \frac{1}{2} \geq \frac{1}{2} - \frac{1}{4\pi}$.

**Claim 4.** *If $j \notin S$ then there exists $w^*$ with $\|w^*\| = \frac{1}{\sqrt{2\pi}}$ such that* $\text{err}_{\mathcal{D}_j}(\text{ReLU}_{w^*}) < \frac{1}{2} - \frac{1}{4\pi} - \frac{2^{-O(k)}}{1 - 2\eta}$.

Claims 3 and 4 imply that we have a gap of at least $\frac{2^{-O(k)}}{1 - 2\eta} = \frac{2^{-ck}}{1 - 2\eta}$ for some $c > 0$ between the relevant and irrelevant variable case. Setting $\epsilon = \frac{2^{-ck}}{1 - 2\eta}$ in Algorithm 1 will let us detect this gap. Since $\mathcal{A}$ is an agnostic learner for ReLU, as long as $M_1 = T(d, 2/\epsilon)$ we know that with probability $2/3$, for all $j \in S$, $\mathcal{A}$ runs on $\mathcal{S}_j$ and outputs $h_j$ such that $\text{err}_{\mathcal{D}_j}(h_j) \leq \min_w \text{err}_{\mathcal{D}_j}(\text{ReLU}_w) \leq \frac{1}{2} - \frac{1}{4\pi} - \epsilon/2$, and for all $j \notin S$, $\text{err}_{\mathcal{D}_j}(h_j) \geq \frac{1}{2} - \frac{1}{4\pi}$.

Using standard concentration inequalities for sub-Gaussian and subexponential random variables [Ver] we see that using a validation set of $M_2 = 100/\epsilon^2$ samples, we have for all $j$, $|\widehat{\text{err}}_{\mathcal{V}_j}(h_j) - \text{err}_{\mathcal{D}_j}(h_j)| \leq \epsilon/4$. Therefore, we can differentiate the two cases as in the Algorithm with confidence $> 1/2$. It is easy to see that the run time of the algorithm is $O(d)T(d, 2/\epsilon) + O(1/\epsilon^2)$, and that this can be amplified to obtain an algorithm with any desired confidence using standard techniques. $\quad\square$

## 4 Lower Bounds for SQ Algorithms

A consequence of Theorem 3 is that any *statistical-query* algorithm for agnostically learning a ReLU with respect to Gaussian marginals yields a statistical-query algorithm for learning parity functions on $k$ unknown input bits. This implies that there is no polynomial time statistical-query (SQ) algorithm that learns a ReLU with respect to Gaussian marginals for a certain restricted class of queries. We present the formal theorem and defer the proof to the supplemental.

**Theorem 4.** *Any SQ algorithm for agnostically learning a ReLU with respect to any distribution $\mathcal{D}$ satisfying Gaussian marginals over the attributes, requires $d^{\Omega(\log(1/\epsilon))}$ unit norm correlation queries or queries independent of the target with tolerance $\frac{1}{\text{poly}(d, 1/\epsilon)}$ to an oracle that returns $\tau$-approximate expectations with respect to $\mathcal{D}$.*

**Remark**: Note that this implies there is no $d^{o(1/\epsilon)}$-time gradient descent algorithm that can agnostically learn $\text{ReLU}(\mathbf{w} \cdot \mathbf{x})$, under the reasonable assumption that for every $i$ the gradients of $\mathbb{E}_{(\mathbf{x}, y) \sim \mathcal{D}} \left[ \left( \text{ReLU}_{\mathbf{w}}(\nu(\mathbf{x})_{-i}) - \frac{y+1}{2} \right)^2 \right]$ can be computed by $O(d)$ queries whose norms are polynomially bounded.

## 5 Approximation Algorithm

In this section we give a learning algorithm that runs in polynomial time in all input parameters and outputs a ReLU that has error $O(\text{opt}^{2/3}) + \epsilon$ where opt is the error of the best-fitting ReLU. The main reduction is a hard thresholding of the labels to create a training set with Boolean labels. We then apply a recent result giving a polynomial-time approximation algorithm for agnostically learning halfspaces over the Gaussian distribution due to Awasthi et. al. [ABL14]. We present our algorithm and give a proof of its correctness.

---
**Algorithm 2**

---
    **Input** Training set $\mathcal{S}$ of $m$ samples $(\mathbf{x}^i, y^i)_{i=1}^m$, the agnostic halfspace learning algorithm
        $\mathcal{A}$ from [ABL14] and a parameter $\alpha$
    **Output** Weight vector $\widehat{\mathbf{w}}$

1: Construct $S' := \{(\mathbf{x}, \mathsf{sign}(y - \alpha)) \mid (\mathbf{x}, y) \in S\}$.
2: Run $\mathcal{A}$ to recover $\widehat{\mathbf{w}}$ close in $\mathsf{err}_{0/1}$.
3: Return $\widehat{\mathbf{w}}$

---

**Theorem 5.** *There is an algorithm (Algorithm 2) that given $O(\mathsf{poly}(d, 1/\epsilon))$ samples $(\boldsymbol{x}, y)$ such that $\boldsymbol{x}$ is drawn from $N(0, I_d)$ and $y \in [0, 1]$ recovers a unit vector $\boldsymbol{w}$ such that $\mathsf{err}(\mathsf{ReLU}_{\boldsymbol{w}}) \leq O(\mathsf{opt}^{2/3}) + \epsilon$ where $\mathsf{opt} := \min_{\|\boldsymbol{w}\|=1} \mathsf{err}(\mathsf{ReLU}_{\boldsymbol{w}})$.*

*Proof.* Let $\mathbf{w}^* = \arg\min_{\|\mathbf{w}\|=1} \mathsf{err}(\mathsf{ReLU}_{\mathbf{w}})$ and so, $\mathsf{err}(\mathsf{ReLU}_{\mathbf{w}^*}) = \mathsf{opt}$. Define the $S_{good}$ to be the set of points that are $\alpha$-close to the optimal ReLU, i.e. $S_{good} = \{x : |y - \mathsf{ReLU}_{\mathbf{w}^*}(\mathbf{x})| \leq \alpha\}$. By Markov's inequality,

$$\Pr[\mathbf{x} \notin S_{good}] = \Pr[|y - \mathsf{ReLU}_{\mathbf{w}^*}(\mathbf{x})| \geq \alpha] \leq \frac{\mathsf{opt}}{\alpha^2}.$$

This implies that all but an $\frac{\mathsf{opt}}{\alpha^2}$ fraction of the points are $\alpha$-close to their corresponding $y$'s. In the first step of Algorithm 2, the labels become Boolean. Define the $0/1$ error of the vector $\mathbf{w}$ as follows, $\mathsf{err}_{0/1}(\mathbf{w}) = \mathbb{E}[\mathsf{sign}(y - \alpha) \neq \mathsf{sign}(\mathbf{w} \cdot \mathbf{x})]$. Let $\mathbf{w}^\dagger$ be the argmin of $\mathsf{err}_{0/1}(\mathbf{w})$ over all vectors $\mathbf{w}$ with $\|\mathbf{w}\|_2 \leq 1$. Since for all elements in $S_{good} \setminus \{\mathbf{v} : \mathbf{w}^* \cdot \mathbf{v} \in (0, 2\alpha)\}$, $\mathsf{sign}(y - \alpha) = \mathsf{sign}(\mathbf{w}^* \cdot \mathbf{x})$,

$$\begin{aligned}
\mathsf{err}_{0/1}(w^*) &\leq \Pr[\mathbf{x} \notin S_{good} \setminus \{\mathbf{v} : \mathbf{w}^* \cdot \mathbf{v} \in (0, 2\alpha)\}] \\
&\leq \Pr[\mathbf{x} \notin S_{good}] + \Pr[\mathbf{x} \in \{\mathbf{v} : \mathbf{w}^* \cdot \mathbf{v} \in (0, 2\alpha)\}] \\
&\leq \frac{\mathsf{opt}}{\alpha^2} + \frac{1}{\sqrt{2\pi}} \int_0^{2\alpha} e^{-g^2/2} dg \;\leq\; \frac{\mathsf{opt}}{\alpha^2} + 2\alpha.
\end{aligned}$$

We now apply Theorem 8 from [ABL14] which gives an algorithm with polynomial running time in $d$ and $1/\epsilon$ that outputs a $\mathbf{w}$ such that $\|\mathbf{w}\| = 1$ and $\|\mathbf{w} - \mathbf{w}^\dagger\| \leq O\left(\left(\frac{\mathsf{opt}}{\alpha^2} + 2\alpha\right)\right) + \epsilon$. For unit vectors $\mathbf{a}, \mathbf{b}$, $\theta(\mathbf{a}, \mathbf{b}) < C \Pr[\mathsf{sign}(\mathbf{a} \cdot \mathbf{x}) \neq \mathsf{sign}(\mathbf{b} \cdot \mathbf{x})]$ for some absolute constant $C$ where $\theta(\mathbf{a}, \mathbf{b})$ is the angle between the vectors (see Lemma 2 in [ABL14]). The triangle inequality and the fact that $\|\mathbf{a} - \mathbf{b}\| \leq \theta(\mathbf{a}, \mathbf{b})$ implies that if $\mathsf{err}_{0/1}(\mathbf{a}), \mathsf{err}_{0/1}(\mathbf{b}) < \eta$ then $\|\mathbf{a} - \mathbf{b}\| \leq C \Pr[\mathsf{sign}(\mathbf{a} \cdot \mathbf{x}) \neq \mathsf{sign}(\mathbf{b} \cdot \mathbf{x})] \leq O(\eta)$. Applying this to $\mathbf{w}^\dagger$ and $\mathbf{w}^*$ yields $\|\mathbf{w}^\dagger - \mathbf{w}^*\| < O(\frac{\mathsf{opt}}{\alpha^2} + 2\alpha)$. Since the ReLU function is 1-Lipschitz, we have

$$\begin{aligned}
\mathsf{err}(\mathsf{ReLU}_{\mathbf{w}}) &= \mathbb{E}[(y - \mathsf{ReLU}(\mathbf{w} \cdot \mathbf{x}))^2] \\
&\leq 2\mathbb{E}[(y - \mathsf{ReLU}(\mathbf{w}^* \cdot \mathbf{x}))^2] + 2\mathbb{E}[(\mathsf{ReLU}(\mathbf{w}^* \cdot \mathbf{x}) - \mathsf{ReLU}(\mathbf{w} \cdot \mathbf{x}))^2] \\
&\leq 2\mathsf{opt} + 2\mathbb{E}[((\mathbf{w}^* - \mathbf{w}) \cdot \mathbf{x})^2] \\
&= 2\mathsf{opt} + 2\|\mathbf{w}^* - \mathbf{w}\|^2 \;\leq\; O\left(\mathsf{opt} + \left(\frac{\mathsf{opt}}{\alpha^2} + 2\alpha\right)^2 + \epsilon\right)
\end{aligned}$$

Setting $\alpha = \mathsf{opt}^{1/3}$ and rescaling $\epsilon$ we have $\mathsf{err}(\mathsf{ReLU}_{\mathbf{w}}) \leq O(\mathsf{opt}^{2/3}) + \epsilon$. $\qquad\square$

## 6  Conclusions and Open Problems

We have shown hardness for solving the empirical risk minimization problem for just one ReLU with respect to Gaussian distributions and given the first nontrivial approximation algorithm. Can we achieve approximation $O(\mathsf{opt}) + \epsilon$? Note our results holds only for the case of unbiased ReLUs, as the constant function $1/2$ may achieve smaller square-loss than any unbiased ReLU. Interestingly, all positive results that we are aware of for learning ReLUs (or one-layer ReLU networks) with respect to Gaussians also assume the ReLU activations are unbiased (e.g., [BG17, Sol17, GKM18, GKLW19, GLM18, ZYWG19]). How difficult is the biased case?

## Acknowledgments

Surbhi Goel and Adam R. Klivans were supported by NSF Award CCF-1717896. Sushrut Karmalkar was supported by NSF Award CNS-1414023.

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
