[Supplementary Material]

# Supplemnental for Time/Accuracy Tradeoffs for Learning a ReLU with respect to Gaussian Marginals

**Surbhi Goel**
Department of Computer Science
University of Texas at Austin
surbhi@cs.utexas.edu

**Sushrut Karmalkar**
Department of Computer Science
University of Texas at Austin
sushrutk@cs.utexas.edu

**Adam Klivans**
Department of Computer Science
University of Texas at Austin
klivans@cs.utexas.edu

## A   Useful Properties

**Fact A** ([2]). *For $\beta_1, \ldots, \beta_k$ such that $\sum_{i=1}^{k} \beta_i^2 = 1$, we have*

$$H_n \left( \sum_{i=1}^{k} \beta_i x_i \right) = \sum_{n_1 + \ldots + n_k = n} \frac{n!}{n_1! \cdots n_k!} \prod_{j=1}^{k} \beta_j^{n_j} H_{n_j}(x_j).$$

**Lemma A.** *For any $\mathbf{w}$, we have*

$$\mathbb{E}_{z \sim \mathcal{N}(0,\mathbf{I}_d)} \left[ \mathsf{ReLU}(\mathbf{w} \cdot z)^2 \right] = \frac{\|\mathbf{w}\|^2}{2} \text{ and } \mathbb{E}_{z \sim \mathcal{N}(0,\mathbf{I}_d)} \left[ \mathsf{ReLU}(\mathbf{w} \cdot z) \right] = \frac{\|\mathbf{w}\|}{\sqrt{2\pi}}.$$

*Proof.* Observe that for $\mathbf{z} \sim \mathcal{N}(0, \mathbf{I}_d)$, $\mathbf{w} \cdot \mathbf{z} \sim \mathcal{N}(0, \|\mathbf{w}\|^2)$, thus we have

$$\begin{aligned}
\mathbb{E}_{\mathbf{z} \sim \mathcal{N}(0,\mathbf{I}_d)} \left[ \mathsf{ReLU}(\mathbf{w} \cdot \mathbf{z})^2 \right] &= \mathbb{E}_{g \sim \mathcal{N}(0,\|\mathbf{w}\|^2)} \left[ \mathsf{ReLU}(g)^2 \right] \\
&= \frac{1}{\sqrt{2\pi}\|\mathbf{w}\|} \int_0^{\infty} g^2 e^{-\frac{g^2}{2\|\mathbf{w}\|^2}} \, dg \\
&= \frac{\|\mathbf{w}\|^2}{2}
\end{aligned}$$

The last follows from observing that the integral is $1/2$ of the variance of a $N(0, \|\mathbf{w}\|^2)$ variable. Similarly, we have

$$\begin{aligned}
\mathbb{E}_{\mathbf{z} \sim \mathcal{N}(0,\mathbf{I}_d)} \left[ \mathsf{ReLU}(\mathbf{w} \cdot \mathbf{z}) \right] &= \mathbb{E}_{g \sim \mathcal{N}(0,\|\mathbf{w}\|^2)} \left[ \mathsf{ReLU}(g) \right] \\
&= \frac{1}{\sqrt{2\pi}\|\mathbf{w}\|} \int_0^{\infty} g e^{-\frac{g^2}{2\|\mathbf{w}\|^2}} \, dg \\
&= \frac{\|\mathbf{w}\|}{\sqrt{2\pi}}.
\end{aligned}$$

Here the last equality follows from standard computation of mean of the absolute value of a Gaussian random variable. □

## B  Background on Statistical Query (SQ) Learning

**SQ Model.**  A SQ algorithm is a learning algorithm that succeeds given only estimates of $\mathbb{E}_{\mathbf{x},y\sim\mathcal{D}}[q(\mathbf{x},y)]$ for query functions of the learner's choosing to an oracle up to a fixed tolerance parameter (see for example [5, 3]). We restrict ourselves to queries that are either *correlation* queries, that is, $\mathbb{E}_{\mathbf{x},y\sim\mathcal{D}}[y \cdot g(\mathbf{x})]$ for any function $g$, or queries that are *independent of the target*, that is, $\mathbb{E}_{\mathbf{x},y\sim\mathcal{D}}[h(\mathbf{x})]$ for any function $h$. For example, the $i^{th}$ coordinate of the gradient with respect to $\mathbf{w}$ of the quantity $(\mathsf{ReLU}_{\mathbf{w}}(\mathbf{x})-y)^2$, i.e. $2\cdot 1_+(\mathbf{w}\cdot\mathbf{x})\cdot(\mathsf{ReLU}_{\mathbf{w}}(\mathbf{x})-y)\cdot\mathbf{x}_i$ can be simulated by a correlation query $g(\mathbf{x}) = -2\cdot 1_+(\mathbf{w}\cdot\mathbf{x})\cdot\mathbf{x}_i$ and query $h(\mathbf{x}) = 2\cdot 1_+(\mathbf{w}\cdot\mathbf{x})\cdot\mathsf{ReLU}_{\mathbf{w}}(\mathbf{x})\cdot\mathbf{x}_i$ independent of the target. Therefore queries with sufficiently small tolerance allow us to simulate gradient descent for the loss function $L(\mathbf{w}) = \mathbb{E}_{\mathbf{x},y\sim\mathcal{D}}[(\mathsf{ReLU}_{\mathbf{w}}(\mathbf{x})-y)^2]$ [1].

**SQ Dimension.**  We define the inner product of two functions $f(\mathbf{x}), g(\mathbf{x})$ with respect to a distribution $\mathcal{D}$ to be $\langle f,g\rangle_{\mathcal{D}} := \mathbb{E}_{\mathbf{x}\sim\mathcal{D}}[f(\mathbf{x})g(\mathbf{x})]$. The norm of a function $f$ is just $\sqrt{\langle f,f\rangle_{\mathcal{D}}}$ and two functions $f \neq g$ are said to be orthogonal if $\langle f,g\rangle_{\mathcal{D}} = 0$. The SQ dimension of a function class $\mathcal{F}$ is the largest number of pairwise orthogonal functions that belong to the function class. The following theorem from [6] gives a lower bound on the number of statistical queries needed to learn the function class in terms of its SQ dimension.

**Theorem A** (Restatement of Theorem from [6]).  *Let $\mathcal{F}$ be a concept class and let $s$ be the SQ dimension of $\mathcal{F}$ with respect to $\mathcal{D}$. Then any learning algorithm that uses tolerance parameter lower bounded by $\tau > 0$ and has access to an oracle that returns $\tau$-approximate expectations (with respect to $\mathcal{D}$) of unit norm correlation queries and queries that are independent of the target, requires at least $(s\tau^2 - 1)/2$ queries.*

## C  Omitted Proofs

*Proof of Claim 1.* For $i \geq 2$,

$$\widehat{\mathsf{ReLU}}_i = \frac{1}{\sqrt{2\pi}}\int_{-\infty}^{\infty}\mathsf{ReLU}(x)H_i(x)e^{-\frac{x^2}{2}}dx$$

$$= \frac{1}{\sqrt{2\pi i!}}\int_0^{\infty}xH_i(x)e^{-\frac{x^2}{2}}dx$$

$$= \frac{1}{\sqrt{2\pi i!}}\int_0^{\infty}(H_{i+1}(x)+iH_{i-1}(x))e^{-\frac{x^2}{2}}dx$$

$$= \frac{1}{\sqrt{2\pi i!}}(H_i(0)+iH_{i-2}(0))$$

Here we used the additional property on the recurrence of $H$, that is, $H_{n+1}(x) = xH_n(x) - nH_{n-1}$. $\qquad\square$

*Proof of Claim 2.* Since $\sum_{i\in S}\left(\frac{1}{\sqrt{k}}\right)^2 = 1$ using Fact A, we have

$$\mathsf{ReLU}\left(\frac{\sum_{i\in S}z_i}{\sqrt{k}}\right) = \sum_{n=0}^{\infty}\frac{\widehat{\mathsf{ReLU}}_n}{\sqrt{n!}}\cdot H_n\left(\frac{\sum_{i\in S}z_i}{\sqrt{k}}\right)$$

$$= \sum_{n=0}^{\infty}\frac{\widehat{\mathsf{ReLU}}_n}{\sqrt{n!}}\cdot\left[\sum_{n_1+\ldots+n_k=n}\frac{n!}{n_1!\cdots n_k!}\prod_{j=1}^{k}\frac{1}{k^{n_j/2}}H_{n_j}(z_j)\right]$$

$$= \sum_{n=0}^{\infty}\frac{\widehat{\mathsf{ReLU}}_n}{\sqrt{n!}}\cdot\left[\frac{1}{k^{n/2}}\cdot\sum_{n_1+\ldots+n_k=n}\frac{n!}{n_1!\cdots n_k!}\prod_{j=1}^{k}H_{n_j}(z_j)\right]$$

$$= \sum_{n=0}^{\infty} \frac{\widehat{\mathrm{ReLU}}_n}{\sqrt{n!}} \cdot \left[ \frac{1}{k^{n/2}} \sum_{n_1+\ldots+n_k=n} \frac{n!}{n_1! \cdots n_k!} \prod_{j=1}^{k} \sqrt{n_j!} \bar{H}_{n_j}(z_j) \right]$$

$$= \sum_{n=0}^{\infty} \frac{\widehat{\mathrm{ReLU}}_n}{k^{n/2}} \cdot \sum_{n_1+\ldots+n_k=n} \left( \frac{n!}{n_1! \cdots n_k!} \right)^{1/2} \prod_{j=1}^{k} \bar{H}_{n_j}(z_j)$$

$\square$

*Proof of Claim 3.* Since $j \in S$, removing index $j$, $\mathbb{E}_{z_j}[\chi_S(\mathbf{z})|\mathbf{z}_{-j}] = 0$. Thus, for the input, the label is a Bernoulli random variable with probability $1/2$. Thus we have,

$$\mathrm{err}_{D_j}(\mathrm{ReLU}_\mathbf{w})$$
$$= \mathbb{E}_{\mathbf{z} \sim \mathcal{N}(0, \mathbf{I}_d)}[(\mathrm{ReLU}(\mathbf{w} \cdot \mathbf{z}_{-j}) - y')^2]$$
$$= \frac{1}{2} \left( \mathbb{E}_{\mathbf{z}_{-j} \sim \mathcal{N}(0, \mathbf{I}_{d-1})} \left[ (\mathrm{ReLU}(\mathbf{w} \cdot \mathbf{z}_{-j}) - 1)^2 \right] + \mathbb{E}_{\mathbf{z}_{-j} \sim \mathcal{N}(0, \mathbf{I}_{d-1})} \left[ (\mathrm{ReLU}(\mathbf{w} \cdot \mathbf{z}_{-j}))^2 \right] \right)$$
$$= \mathbb{E}_{\mathbf{z}_{-j} \sim \mathcal{N}(0, \mathbf{I}_{d-1})} \left[ \mathrm{ReLU}(\mathbf{w} \cdot \mathbf{z}_{-j})^2 \right] - \mathbb{E}_{\mathbf{z}_{-j} \sim \mathcal{N}(0, \mathbf{I}_{d-1})} \left[ \mathrm{ReLU}(\mathbf{w} \cdot \mathbf{z}_{-j}) \right] + \frac{1}{2}$$
$$= \frac{||\mathbf{w}||^2}{2} - \frac{||\mathbf{w}||}{\sqrt{2\pi}} + \frac{1}{2}$$

Here the third equality follows since $j \in S$ and not in $\mathbf{z}_{-j}$ therefore, the label is random for the ReLU. The last equality follows from Lemma A. Note that, for any ReLU, the minimum error is achieved when $||\mathbf{w}|| = \frac{1}{\sqrt{2\pi}}$. Thus when $j \notin S$ the best ReLU achieves error at least $\frac{1}{2} - \frac{1}{4\pi}$. $\square$

*Proof of Claim 4.* Since $j$ is not a relevant variable $S \subseteq [d] \setminus \{j\}$, from Theorem 1, we know that there exists $\mathrm{ReLU}_{\mathbf{w}_S}$ with $||\mathbf{w}_S|| = 1/\sqrt{2\pi}$ dependent only on variables in $S$ correlated with $\chi_S^\gamma$,

$$\mathrm{err}_{D_j}(\mathrm{ReLU}_{\mathbf{w}_S})$$
$$= \mathbb{E}_{\mathbf{z} \sim \mathcal{N}(0, \mathbf{I}_d)}[(\mathrm{ReLU}(\mathbf{w}_S \cdot \mathbf{z}) - y')^2]$$
$$= (1 - \eta) \mathbb{E}_{\mathbf{z} \sim \mathcal{N}(0, \mathbf{I}_d)} \left[ \left( \mathrm{ReLU}(\mathbf{w}_S \cdot \mathbf{z}) - \frac{\prod_{i \in S} \mathrm{sign}(z_i) + 1}{2} \right)^2 \right]$$
$$\quad + \eta \mathbb{E}_{\mathbf{z} \sim \mathcal{N}(0, \mathbf{I}_d)} \left[ \left( \mathrm{ReLU}(\mathbf{w}_S \cdot \mathbf{z}) - \frac{1 - \prod_{i \in S} \mathrm{sign}(z_i)}{2} \right)^2 \right]$$
$$= \mathbb{E}_{\mathbf{z} \sim \mathcal{N}(0, \mathbf{I}_d)} \left[ \mathrm{ReLU}(\mathbf{w}_S \cdot \mathbf{z}_S)^2 \right] - (1 - 2\eta) \mathbb{E}_{\mathbf{z} \sim \mathcal{N}(0, \mathbf{I}_d)} \left[ \mathrm{ReLU}(\mathbf{w}_S \cdot \mathbf{z}) \prod_{i \in S} \mathrm{sign}(z_i) \right]$$
$$\quad - \mathbb{E}_{\mathbf{z} \sim \mathcal{N}(0, \mathbf{I})} \left[ \mathrm{ReLU}(\mathbf{w}_S \cdot \mathbf{z}) \right] + \frac{1}{2} \mathbb{E}_{\mathbf{z} \sim \mathcal{N}(0, \mathbf{I}_d)} \left[ \prod_{i \in S} \mathrm{sign}(z_i)^2 \right]$$
$$= \frac{||\mathbf{w}_S||^2}{2} - \frac{||\mathbf{w}_S||}{\sqrt{2\pi}} + \frac{1}{2} - (1 - 2\eta) \mathbb{E}_{\mathbf{z} \sim \mathcal{N}(0, \mathbf{I}_d)} \left[ \mathrm{ReLU}(\mathbf{w}_S \cdot \mathbf{z}) \prod_{i \in S} \mathrm{sign}(z_i) \right]$$
$$\geq \frac{||\mathbf{w}^*||^2}{2} - \frac{||\mathbf{w}^*||}{\sqrt{2\pi}} + \frac{1}{2} - \frac{2^{-O(k)}}{1 - 2\eta} = \frac{1}{2} - \frac{1}{4\pi} - \frac{2^{-O(k)}}{1 - 2\eta}$$

$\square$

*Proof of Theorem 4.* Define the problem of 'restricted $k$-sparse parities' as the problem of learning an unknown parity function $\chi_S$ over set $S$, where $S$ contains $k$ out of the first $d$ variables over $\mathcal{D}$ with input distribution $\mathcal{D}_b \times \mathcal{N}(0, I_d)_+$. Here $\mathcal{D}_b$ is the uniform distribution on $\{\pm 1\}^d$ and the labels $y(\mathbf{x})$ are given by $\chi_S(\mathbf{x})$. It is easy to see that Theorem A implies that we require $d^{\Omega(k)}$ unit norm queries to learn this function class from queries to an oracle $\mathcal{O}$ with tolerance $1/\mathrm{poly}(d, 2^k)$.

We give a proof by contradiction. Suppose we can agnostically learn ReLUs with respect to Gaussian marginals using an SQ algorithm $\mathcal{A}$ with $d^{o(\log \frac{1}{\epsilon})}$ queries to the corresponding oracle with tolerance $1/\mathrm{poly}(d, \frac{1}{\epsilon})$. We will show how to use $\mathcal{A}$ to design an SQ algorithm for the problem of learning restricted $k$-sparse parities using $d^{o(k)}$ queries contradicting Theorem A.

We essentially use the same approach as in Theorem 3. In order to learn the parity function, the reduction requires us to use an SQ learner to solve $d$ different ReLU regression problems. Consider the mapping $\nu$ that takes as input $\mathbf{x} = (x_1, \ldots x_d, g_1, \ldots, g_d)$ and returns $\mathbf{x}' = (x_1 g_1, \ldots, x_d g_d)$. Using this transformation, the ReLU regression problems we need to minimize are $\mathbb{E}_{(\mathbf{x}, y) \sim \mathcal{D}} \left[ \left( \mathrm{ReLU}_{\mathbf{w}}(\nu(\mathbf{x})_{-i}) - \frac{y+1}{2} \right)^2 \right]$ up to an additive $\epsilon = 2^{-ck}$. Observe that for $(\mathbf{x}, y) \sim \mathcal{D}$, $(\mathbf{x}', y') = (\nu(\mathbf{x}), (y+1)/2)$ is distributed according to some $\mathcal{D}'$ where the distribution on $\nu(\mathbf{x})$ is $\mathcal{N}(0, I_d)$. Thus we can use $\mathcal{A}$ on this distribution to solve the optimization problem. However, in order to run $\mathcal{A}$, we need to simulate the queries $\mathcal{A}$ asks its oracle using $\mathcal{O}$. To do so, for any correlation query function $g$ that $\mathcal{A}$ chooses, that is, query $\mathbb{E}_{(\mathbf{x}', y') \sim \mathcal{D}'} [g(\mathbf{x}') \cdot y']$ we use correlation query function $g' = \frac{1}{2} g \circ \nu$ to $\mathcal{O}$ and for any query $h$ that $\mathcal{A}$ chooses that is independent of the target, we query function $h' = \frac{1}{2} h \circ \nu$.

Since for $k = \Theta(\log \frac{1}{\epsilon})$, such an algorithm would solve the problem of 'restricted $k$-sparse parities' using $d^{o(k)}$ queries of tolerance $1/\mathrm{poly}(d, 2^k)$. This contradicts the $d^{\Omega(k)}$ lower bound on the number of queries required to solve $k$-SPLN of tolerance $1/\mathrm{poly}(d, 2^k)$ we get from Theorem A.

$\square$

*Proof of Theorem 5.* Let $\mathbf{w}^* = \arg \min_{\|\mathbf{w}\|=1} \mathrm{err}(\mathrm{ReLU}_{\mathbf{w}})$ and so, $\mathrm{err}(\mathrm{ReLU}_{\mathbf{w}^*}) = \mathrm{opt}$. Define the $S_{good}$ to be the set of points that are $\alpha$-close to the optimal ReLU, i.e. $S_{good} = \{x : |y - \mathrm{ReLU}_{\mathbf{w}^*}(\mathbf{x})| \leq \alpha\}$. By Markov's inequality,

$$\Pr[\mathbf{x} \notin S_{good}] = \Pr[|y - \mathrm{ReLU}_{\mathbf{w}^*}(\mathbf{x})| \geq \alpha] \leq \frac{\mathrm{opt}}{\alpha^2}.$$

This implies that all but an $\frac{\mathrm{opt}}{\alpha^2}$ fraction of the points are $\alpha$-close to their corresponding $y$'s. In the first step of Algorithm 2, the labels become Boolean. Define the $0/1$ error of the vector $\mathbf{w}$ as follows, $\mathrm{err}_{0/1}(\mathbf{w}) = \mathbb{E}[\mathrm{sign}(y - \alpha) \neq \mathrm{sign}(\mathbf{w} \cdot \mathbf{x})]$. Let $\mathbf{w}^\dagger$ be the argmin of $\mathrm{err}_{0/1}(\mathbf{w})$ over all vectors $\mathbf{w}$ with $\|\mathbf{w}\|_2 \leq 1$. Since for all elements in $S_{good} \setminus \{\mathbf{v} : \mathbf{w}^* \cdot \mathbf{v} \in (0, 2\alpha)\}$, $\mathrm{sign}(y - \alpha) = \mathrm{sign}(\mathbf{w}^* \cdot \mathbf{x})$,

$$\begin{aligned}
\mathrm{err}_{0/1}(w^*) &\leq \Pr[\mathbf{x} \notin S_{good} \setminus \{\mathbf{v} : \mathbf{w}^* \cdot \mathbf{v} \in (0, 2\alpha)\}] \\
&\leq \Pr[\mathbf{x} \notin S_{good}] + \Pr[\mathbf{x} \in \{\mathbf{v} : \mathbf{w}^* \cdot \mathbf{v} \in (0, 2\alpha)\}] \\
&\leq \frac{\mathrm{opt}}{\alpha^2} + \frac{1}{\sqrt{2\pi}} \int_0^{2\alpha} e^{-g^2/2} dg \ \leq \ \frac{\mathrm{opt}}{\alpha^2} + 2\alpha.
\end{aligned}$$

We now apply Theorem 8 from [1] which gives an algorithm with polynomial running time in $d$ and $1/\epsilon$ that outputs a $\mathbf{w}$ such that $\|\mathbf{w}\| = 1$ and $\|\mathbf{w} - \mathbf{w}^\dagger\| \leq O((\frac{\mathrm{opt}}{\alpha^2} + 2\alpha)) + \epsilon$. For unit vectors $\mathbf{a}, \mathbf{b}$, $\theta(\mathbf{a}, \mathbf{b}) < C \Pr[\mathrm{sign}(\mathbf{a} \cdot \mathbf{x}) \neq \mathrm{sign}(\mathbf{b} \cdot \mathbf{x})]$ for some absolute constant $C$ where $\theta(\mathbf{a}, \mathbf{b})$ is the angle between the vectors (see Lemma 2 in [1]). The triangle inequality and the fact that $\|\mathbf{a} - \mathbf{b}\| \leq \theta(\mathbf{a}, \mathbf{b})$ implies that if $\mathrm{err}_{0/1}(\mathbf{a}), \mathrm{err}_{0/1}(\mathbf{b}) < \eta$ then $\|\mathbf{a} - \mathbf{b}\| \leq C \Pr[\mathrm{sign}(\mathbf{a} \cdot \mathbf{x}) \neq \mathrm{sign}(\mathbf{b} \cdot \mathbf{x})] \leq O(\eta)$. Applying this to $\mathbf{w}^\dagger$ and $\mathbf{w}^*$ yields $\|\mathbf{w}^\dagger - \mathbf{w}^*\| < O(\frac{\mathrm{opt}}{\alpha^2} + 2\alpha)$. Since the ReLU function is 1-Lipschitz, we have

$$\begin{aligned}
\mathrm{err}(\mathrm{ReLU}_{\mathbf{w}}) &= \mathbb{E}[(y - \mathrm{ReLU}(\mathbf{w} \cdot \mathbf{x}))^2] \\
&\leq 2\mathbb{E}[(y - \mathrm{ReLU}(\mathbf{w}^* \cdot \mathbf{x}))^2] + 2\mathbb{E}[(\mathrm{ReLU}(\mathbf{w}^* \cdot \mathbf{x}) - \mathrm{ReLU}(\mathbf{w} \cdot \mathbf{x}))^2] \\
&\leq 2\mathrm{opt} + 2\mathbb{E}[((\mathbf{w}^* - \mathbf{w}) \cdot \mathbf{x})^2] \\
&= 2\mathrm{opt} + 2\|\mathbf{w}^* - \mathbf{w}\|^2 \ \leq \ O\left( \mathrm{opt} + \left( \frac{\mathrm{opt}}{\alpha^2} + 2\alpha \right)^2 + \epsilon \right)
\end{aligned}$$

Setting $\alpha = \mathrm{opt}^{1/3}$ and rescaling $\epsilon$ we have $\mathrm{err}(\mathrm{ReLU}_{\mathbf{w}}) \leq O(\mathrm{opt}^{2/3}) + \epsilon$. $\square$

## Footnotes

[1] Feldman et. al. [4] have shown that a broad class of first order convex optimization methods including gradient descent – but excluding stochastic gradient descent – can be simulated using statistical queries