[Reviews · NeurIPS 2019]

Reviewer 1



This paper show that for the task of learning the best possible ReLU to fit some labeled data (whose unlabeled points are from a spherical Gaussian) (i) assuming hardness of the (well-studied) "Learning Parities with Noise" problem, no algorithm can have error OPT + o(1) (where OPT is the error of the best ReLU fit) in polynomial time (in the dimension n) (ui) unconditionally, no SQ (statistical query; includes in particular gradient descent and most practically used algorithms) algorithm can have error OPT + o(1) (where OPT is the error of the best ReLU fit) in polynomial time (in the dimension n) The authors also show that, while OPT+o(1) is not achievable, it is possible to get error OPT^{2/3} + o(1) in polynomial time. Both upper and lower bound proceed by (careful) reductions, one to the Parity with Noise problem, the other to the problem of (agnostic) learning linear threshold functions in the Gaussian space. The paper is very clear, and though I am not an expert I found it interesting and well written. I recommend acceptance, modulo the few comments below. ========== - ll. 38-41: this is confusing, as "Corollary 1" is not a corollary of Theorem 1. I strongly recommend you detail a bit these two lines to make this clear: Theorem 1 and Corollary 1 both rely on the same reduction, but Theorem 1 (for all algorithms) relies on Assumption 1 as base case, while Corollary 1 (for SQ algorithms only) is unconditional based on the hardness of learning parity with noise for SQ algorithms. - everywhere: "hermite" needs to be capitalized, it's a name - ll. 266-267: I didn't quite get the distinction and short discussion about biased ReLU. Can you clarify? UPDATE: I have read the author's response, and am satisfied with it.

Reviewer 2



The paper investigates the complexity of learning a ReLU with noise and under the Gaussian distribution, which arguably is a more interesting case than at least some of the previously considered ones (in which there was no noise or on the lower bound side of things, the hardness steamed from the underlying distribution). Assuming the hardness of learning sparse parities with noise, the paper shows that getting an additive epsilon approximation requires n^Omega(log(1/eps)) time. The reduction also proves that a gradient descent algorithm cannot converge in polynomial time. The second main result is an efficient algorithm for learning an ReLU with polynomially larger error than optimal (opt^(2/3) instead of opt). This result is obtained by exploiting a connection to learning halfspaces. I think both the considered setting (learning ReLUs in the Gaussian setting with noise) and connections to learning parities and halfspaces are very appealing. I therefore vote for accepting the paper.

Reviewer 3



This paper studies the computational complexity of learning a single ReLU with respect to Gaussian examples. Since ReLUs are now the standard choice of nonlinearity in deep neural networks, the computational complexity of learning them is clearly of interest. Of course, the computational complexity of learning a ReLU may depend substantially on the specific setting assumed; it is interesting to understand the range of such assumptions and their implications for complexity. This paper studies the following setting: given independent samples (x_1,y_1), ... , (x_n,y_n) where x is spherical Gaussian in d dimensions and y \in R is arbitrary, find a ReLU function f_w(x) = max(0, w \cdot x) for some vector w with minimal mean squared error \sum (y_i - f(x_i))^2. (This is agnostic learning since the y's are arbitrary.) The main results are as follows: 1) There is no algorithm to learn a single ReLU with respect to Gaussian examples to additive error \epsilon in time d^{o(log 1/\epsilon)} unless $k$-sparse parities with noise can be learned in time d^{o(k)} 2) If opt = min_{w} (mean squared error of f) then (with normalization such that opt \in [0,1]) there is an algorithm which agnostically learns a ReLU to error opt^{2/3} + \epsilon in time poly(d,1/\epsilon). The proof of (1) goes via Hermite analysis (i.e. harmonic/Fourier analysis in Gaussian space) together with a nice coordinate deletion'' trick to use a ReLU learner to identify the relevant coordinates in a k-sparse parity with noise. The proof of (2) uses a reduction to a recent algorithm for agnostically learning halfspaces. An important feature of (1) is the contrast it presents to recent results showing that gradient descent can learn ReLUs in polynomial time *when the labels y_i are exactly equal to some ReLU* -- i.e. in the realizable setting. I am not an expert in all the prior work, this paper appears to clear both the novelty and importance bars for acceptance to NeurIPS. The paper is reasonably well-written. I would prefer if the authors were somewhat more formal in their theorem statements in the introduction, and gave a more detailed SQ lower bound (as it is there is some lack of rigor to support the rather bold claim in the introduction about gradient descent). It would also be good to point out that the difference (as I understand it) between the running times ruled out by the lower bound and what the approximation algorithm achieves is d^(log(1/epsilon)) versus poly(d,1/epsilon) -- in many contexts these two running times are not too dissimilar, at least as compared to, say, d^(1/epsilon). (Is there an obvious d^(1/epsilon) time algorithm, for instance?) UPDATE: I have read the author responses. I agree it is important to rule out any algorithm with fixed polynomial running time, and that the lower bound is quite interesting. I think it would be nice point out in the final paper that the complexity of the problem remains fairly wide open -- e.g. it might be possible to give a either a lower bound or algorithm for time $d^(1/\epsilon)$. I continue to feel that the paper should be accepted.

[Author Response · NeurIPS 2019]

We thank the reviewers for their comments and suggestions. We address the specific the specific concerns below.

**Reviewer 1:** - ll. 38-41: We agree that Corollary 1 is not a direct corollary. It does, however, follow from the
techniques used to prove Theorem 1, and the details of this are described in Section 3.1. We will state and prove this
more formally in the final version.

- ll. 266-267: We were trying to point out that for example for random Boolean functions, the constant hypothesis 1/2
achieves smaller square loss than any homogeneous ReLU.

**Reviewer 2:** Thank you for pointing out the typo. We will fix it.

**Reviewer 3:** The details of the SQ lower bound are mentioned in section 3.1. We will be more precise in our theorem
statements in the final version.

Regarding the $d^{\Omega(\log(1/\epsilon))}$ lower bound, note that this rules out any algorithm that has a running time that is a fixed
polynomial in the dimension. This is stronger than, say a $(1/\epsilon)^{\log(1/\epsilon)}$ lower bound. We are not aware of a $d^{O(1/\epsilon)}$ time
algorithm for this problem.

[Meta-Review · NeurIPS 2019]

The authors achieve three strong results on a fundamental problems of central interest.